# Wondrous Yellow Molecule: Are Hydrogels a Successful Strategy to Overcome the Limitations of Curcumin?

**DOI:** 10.3390/molecules29081757

**Published:** 2024-04-12

**Authors:** Magdalena Stachowiak, Dariusz T. Mlynarczyk, Jolanta Dlugaszewska

**Affiliations:** 1Chair and Department of Chemical Technology of Drugs, Poznan University of Medical Sciences, Rokietnicka 3, 60-806 Poznan, Poland; 2Chair and Department of Genetics and Pharmaceutical Microbiology, Poznan University of Medical Sciences, Rokietnicka 3, 60-806 Poznan, Poland

**Keywords:** bioavailability, curcumin, formulations, hydrogel, stability

## Abstract

Curcumin is a natural compound with a great pharmaceutical potential that involves anticancer, anti-inflammatory, antioxidant, and neuroprotective activity. Unfortunately, its low bioavailability, instability, and poor water solubility significantly deteriorate its clinical use. Many attempts have been made to overcome this issue, and encapsulating curcumin in a hydrogel matrix may improve those properties. Hydrogel formulation is used in many drug delivery forms, including classic types and novel forms such as self-assembly systems or responsive to external factors. Reviewed studies confirmed better properties of hydrogel-stabilized curcumin in comparison to pure compound. The main enhanced characteristics were chemical stability, bioavailability, and water solubility, which enabled these systems to be tested for various diseases. These formulations were evaluated for wound healing properties, effectiveness in treating skin diseases, and anticancer and regenerative activity. Hydrogel formulation significantly improved biopharmaceutical properties, opening the opportunity to finally see curcumin as a clinically approved substance and unravel its therapeutic potential.

## 1. Introduction

### 1.1. Curcumin

Curcumin is a natural compound derived mainly from the rhizome of *Curcuma longa* L. [1,2,3,4,5]. It presents a wide range of biological activities to which mainly three reactive sites in its chemical structure contribute (Figure 1) [2,6]. Each site is responsible for different types of interactions. The hydroxyl group acts as a hydrogen donor and enables sweeping of free radicals. Additionally, α,β-unsaturated β-diketone moiety is engaged in nucleophilic addition reaction as a Michael acceptor and is capable of chelating metal ions [3,6,7].

The pharmaceutical potential of curcumin is proven in numerous studies. Curcumin exhibits activity in various stages against many types of cancer [2,4,7,8]. Other than that, it presents, among others, anti-inflammatory, antioxidant, and neuroprotective effects, which may be useful in treating many diseases, such as autoimmune, neurodegenerative, or inflammatory diseases, including Alzheimer’s, osteoarthritis, lupus, and multiple sclerosis [2,3,7,8,9,10]. When incorporated into textile wound dressings, it shows potential in wound healing [11]. The antibacterial, antiviral, and antifungal activity of curcumin have also been scientifically proved [2,7,12,13,14].

Curcumin’s broad potential is attributed to its influence on many molecular pathways. It has an impact on many transcription factors, such as NFκB, which results in anticancer, anti-inflammatory, antioxidant, neuro-, and chemoprotective properties. Studies showed that modulating NFκB also contributes to anticancerogenic properties due to suppressing TNF-α inflammatory response. Curcumin was shown to inhibit NFκB in many ways, such as upregulating transcription factor NRf2 and TRL4 receptors, increasing adiponectin production, or suppressing STAT proteins [1,5,6,8,9,10,13]. Other than that, curcumin enhances endothelial heme oxygenase-1 activity, which improves its endurance for reactive oxygen species (ROS) and inhibits cyclo- and lipoxygenase (responsible for generating ROS) as well as COX-I and COX-II (engaged in inflammatory response) [3,14]. There are also many biochemical pathways associated with cancers that curcumin affects (Figure 2), including caspase activation pathways (caspase-3, caspase-7, caspase-8, caspase-9), cell proliferation, metastasis and survival pathways (MMP-9, MMP-2, ICAM-1, CXCR, COX-II, cyclin D1,c-MYC, IL-6, IL-8, IAP-1, Bcl-2, Bcl-xl, cFLIP, XIAP), death receptor pathways (DR4, DR5), protein kinase pathways (PKA, PhK, MAPK, AMPK, AKT, JNK), and tumor suppressor pathways (p53, p21) [4,6,8,15]. Curcumin affects multiple more signaling pathways other than the cancer-related ones, which contribute, among others, to the destabilization of FtsZ ring formation and interference proliferation of Gram-negative bacteria (such as *E*. *coli* and *P*. *aeruginosa*) [1,12]. Another effect exerted by curcumin includes the inhibition of Aβ protein aggregation and its removal, which leads to a decrease in the amount of Aβ accumulation in the brain [3,15]. Curcumin also contributes to epigenetic modifications, such as histone alterations, DNA methylation, and micro-RNA expression. For example, curcumin inhibits HDAC2 activity, thus reducing the severity of chronic obstructive pulmonary disease (COPD). HDAC suppression may be also associated with nerve regeneration. Curcumin also inhibits heteromeric amino acid transporters (HAT) activity in rats, resulting in protective activity against heart failure and neuropathic pain [15,16].

Moreover, many studies have proven that curcumin is safe for human intake [8,13,15]. It is GRAS (“generally recognized as safe”) by the FDA (“Food and Drug Administration”). JECFA (“Joint Food and Agriculture Organization/World Health Organization Expert Committee on Food Additives”) and EFSA (“European Food Safety Authority”) established an allowable daily intake to 0–3 mg/kg of body weight. Clinical trials confirmed its safety in doses up to 12 g/day [2,3,5,6,8,10,15,17]. Unfortunately, the high level of safety profile can be associated with its poor ability to achieve significant concentrations in blood due to its chemical and metabolic instability and weak bioavailability. It belongs to class IV in the Biopharmaceutical Classification System. It exhibits poor water-solubility (due to its hydrophobicity) and undergoes rapid degradation, especially in physiological and higher pH. Combining the above characteristics with curcumin’s poor ADME (Absorption, Distribution, Metabolism, Excretion) properties strongly limits the pharmaceutical utility of the compound. Low bioavailability after oral intake is caused by relatively low absorption in the small intestine and quick liver metabolism (it is metabolized by 1st and 2nd phase enzymes), followed by rapid elimination in the gallbladder. Studies show that after oral intake, even in high doses (up to 12 g/day), the curcumin levels in blood were insignificant, up to ng/L. Research confirmed the lack of specific distribution on a significant level [2,3,7,10,15,18].

However, attempts are being made to exploit this molecule’s potent biological activity. Most common are the studies concerning various chemical modifications and pharmaceutical formulations aiming to increase the bioavailability of this compound [10,14,15,17]. Some of them concern synthesizing curcumin analogues through structure modification; however, the main focus is put on novel curcumin formulations. The curcumin nanoformulations contribute to solubility enhancement, stabilization of the compound, and increased bioavailability. Research showed that the novel nanoparticle curcumin formulation—Theracurmin—presents much higher bioavailability than curcumin itself. Nanoparticles of curcumin were investigated in many studies. It was proven that this formulation retains the biological activity of curcumin and presents antibacterial, immunomodulatory, antiviral, and antioxidant properties. Another type of curcumin formulation is a broadly researched phytosomal formulation—Meriva. It exhibited better pharmacokinetic and stronger biological properties than the carried compound by itself. It showed activity against many health conditions, such as diabetes, osteoarthritis, or solid tumors. Interesting and very promising curcumin formulations are hydrogels. Their activity was proven in different studies as thermosensitive hydrogels or hydrogel-functionalized bandages [2,4,5,6,8,10,11,14,17].

### 1.2. Hydrogels

Hydrogels are polymeric materials with a three-dimensional structure (Figure 3). They are hydrophilic and possess the ability to swell and absorb water in significant amounts reaching a thousand times their dry mass [19,20]. The high biocompatibility and similarity of hydrogel to extracellular matrix make them perfect candidates in medical applications [20]. Furthermore, with great biocompatibility and mechanical properties, they also exhibit antimicrobial, antiviral, and antifungal activities [21]. Hydrogels can be divided into two main groups based on their origin—natural and synthetic hydrogels. They are characterized by different properties and each group has their advantages and limitations [20]. Natural hydrogels, such as collagen, chitosan, and alginate, are superior in terms of the biological aspects, while synthetic ones are more stable, characterized by better mechanical properties [19]. One of the biggest assets of hydrogels is their tunability. They respond differently to various stimuli, so their properties can be easily modified by changing synthetic approaches and substrates used [21,22]. Hydrogels are researched for potential use in bioengineering for tissue regeneration: cartilage, bones, muscle, skin, or vascular [23,24]. Currently, there are many pharmaceutical applications for hydrogels. The huge potential of hydrogels lies in their ability to incorporate different substances into their matrix. They are being researched with a good outcome for use as biosensors or non-viral gene-delivery agents in many therapies, including cancer. The advantage of hydrogels over viral vectors used in gene delivery is the lack of immunogenicity. Another aspect of pharmaceutical hydrogel use is drug delivery. They encapsulate the drug in their matrix, which allows the administration of the drug orally, topically, or parenterally, with specific, targeted properties with controlled or delayed release. This formulation increases the bioavailability of hydrophobic drugs and enables combining several compounds in one hydrogel matrix [25,26,27,28]. Considering the above-discussed properties of hydrogels, these matrices can be used as a promising delivery system for low soluble hydrophobic therapeutic agents such as curcumin.

## 2. Use of Hydrogels for Curcumin Delivery

Many studies have been conducted concerning novel curcumin–hydrogel delivery systems. They are designed to take advantage of curcumin’s versatile pharmaceutical potential and hydrogel specific properties while overcoming curcumin’s disadvantages such as poor bioavailability and instability (Table 1) [29,30,31]. Many known and novel delivery systems utilizing various releasing modes were used to create a hydrogel base for the curcumin.

### 2.1. Hydrogels’ Releasing Modes for Curcumin Delivery

One advantage of hydrogels is the prolonged release profile of the carried substance. Such an example might be nanocellulose/chitosan hydrogel (CNC-chitosan hydrogel). By incorporating curcumin into this hydrogel, curcumin’s stability was improved, and its retention time in an acidic medium was extended, both of which contributed to a prolonged release profile after oral administration [30]. Other reported oral delivery systems enabling modified curcumin release from the hydrogel matrix were soy protein/sodium alginate nanogel-based cress seed gum hydrogel [32], 2-hydroxyethyl methacrylate/gelatin/alginate/iron(III) oxide-based hydrogel [36], or poly(*N*-isopropylacrylamide) hydrogel [37].

Hydrogels were also evaluated for the prolonged release profile of curcumin as topical drug delivery systems. The biggest challenge in the topical delivery of curcumin is its insolubility [31,33]. Hence, curcumin nanocrystal-loaded clay-based hydrogels were prepared by using the wet ball media milling method and mixing formed nanosuspension with clay-based hydrogel. Subsequently, the product was analyzed for its topical use. The results of ex vivo experiments indicated improved porcine skin model permeability (with permeation ranging around 40% through the stratum corneum and around 25% through the epidermis) and free radical neutralization (ranging from 40% to 80% depending on the type of the clay) [31]. Similar results were obtained when curcumin nanocrystals were coated with chitosan and further embedded in Carbopol934 hydrogel. Such a system was characterized by enhanced physical and chemical properties, including better skin penetration (from 556 µg for free curcumin suspension to 1798 µg for nanocurcumin hydrogel) and active compound stability [33].

Injectable hydrogels are a good way to deliver the drugs to specific locations—the deep and shallow areas. Depending on hydrogel matrix substrates, injectable drug delivery systems can serve different functions and release curcumin in the desired way [34,38]. Thiolated curcumin-loaded chitosan/poly(ethylene oxide) diacrylate (TCS/PEGDA) injectable hydrogel was synthesized by mixing TCS/PEGDA with curcumin and evaluated both in vitro and in vivo. The drug release profile was found to be stable and controlled. Curcumin was released in a prolonged way, which contributed to higher efficiency. The system was reported to be non-toxic while exhibiting antitumor activity (tinhibition was over 40%) [34]. Another example of hydrogel with controlled drug release in the injectable form is oxidized cyclodextrin-functionalized injectable gelatin hydrogels, which have suitable physical, chemical, and biological properties confirmed in the in vitro studies. Its significant bioadhesive function (2-4-fold higher than fibrin glue) shows great promise in constricting the area of carried substance effect [35].

Alternative use of hydrogel was focused on developing thermosensitive hydrogels as mucoadhesive intranasal delivery systems for brain targeting. Such systems were found to undergo gel–sol transition when applied to the nasal mucosa. Due to its properties and potential to cross the blood–brain barrier, it may help to treat depression [39,40].

Functionalized hydrogels may be designed in such a way that they release the carried compound in response to a specific stimulus. β-Cyclodextrin/curcumin incorporated in chitosan/gellan gum hydrogels were synthesized by consecutively mixing the reagents in a required order. Obtained hydrogels were reported to be both pH- and thermoresponsive. The system was stable between pH 4.9 and 5; the pH values below 4.9 charged hydrogel positive and above 5.6- negative. Depending on the chitosan/gellan gum ratio, the gelation of the hydrogels occurred between 38 °C and 42 °C (with time ranges between 1.4 and 1.5 min). The curcumin in vitro release profile showed its potential in use for gastrointestinal delivery systems [41]. Dual redox- and thermo-responsive properties were also presented by a delivery system prepared by using Pluronic 127 (thermo-sensitive component) and keratin (redox-sensitive component), and their conjugation followed by incorporation of curcumin using a single-step nanoprecipitation technique. During the drug release studies, 70% of the drug was released in the presence of a reducing agent (glutathione), while without it, only 10% of curcumin was released. The results showed good biocompatibility and cellular intake and confirmed the designed dual-sensitive properties [42].

An interesting approach to the targeted release of curcumin from hydrogel formulation is the design of enzyme-triggered release systems. This approach was experimentally tested when curcumin was entrapped in the phenoxyalkylmaleate-based amphiphilic hydrogel. It was proven that curcumin could be released from the hydrogel structure as a result of its exposure to the Lipozyme (curcumin was being released from the hydrogel during the 14 h after enzyme addition due to the hydrogel degradation) or in the acidic pH (the hydrogel remained stable at pH = 8; however, with acid addition and pH below eight, the degradation of the hydrogel and curcumin release were observed) [43].

A different factor may also be a trigger for drug delivery—electrical impulse. Hydrogel delivery systems based on poly(3,4-ethylenedioxythiophene) (PEDOT) release the carried substance in response to electrical current. A study showed controlled release of curcumin from the PEDOT/alginate system when negative voltage was applied (−1.0 V). After 15 min the release was 2-fold higher in comparison to the positive charge. After 2 hours of elecrostimulation, ~25% of curcumin was released, which was 22% higher as compared to non-stimulated experiment after 9 days of simple diffusion and 19% higher than in the positive charge tests, what may suggest its future use in drug delivery implants [44].

So far, the curcumin hydrogel formulations have been evaluated for their protective and regenerative properties, including wound healing, and assessed for treating skin diseases and neoplasms and managing Alzheimer’s disease.

### 2.2. Wound Healing

One of the most promising potential uses of the curcumin–hydrogel combination is its application in wound-healing dressings. Thanks to the unique features of hydrogels and curcumin, such as antibacterial, anti-inflammatory, and antifungal properties, wound dressings create a suitable environment for this process (Figure 4) [11,45,46,47,48].

There are numerous studies concerning this idea. Some approaches involve the modification of traditional bandages with curcumin-loaded hydrogel. One of the research studies used curcumin-containing Z-Tyr-Phe-OH-based hydrogel. Obtained hydrogel was incorporated into cotton and viscose/polyester dressings through impregnation. The results showed improved absorption (swelling increased from ~185% to ~355% for hydrogel-modified bandages and from ~150% to ~355% and ~450% for non-woven), thermal stability (bandages degradation improved from 365 °C to 375 °C; for non-woven degradation it improved from ~380 °C to 438 °C) with maintaining the same level of breathability. The prepared composite gained fungi and bacteriostatic properties compared to the traditional bandages. Additionally, the authors state that the properties of the newly developed wound-dressing system could enhance cell proliferation and wound infection prevention and speed up the healing process compared to conventional dressings [11].

Often, a hydrogel is modified not only with curcumin, but also with other compounds that improve the effectiveness of the material in various ways, such as enhancement of the bactericidal and antioxidant effects. A hydrogel was created by immersing bacterial cellulose into curcumin-incorporated hydroxypropyl-β-cyclodextrin and tested for its wound treatment applications. The developed system created a suitable environment for wound treatment and exhibited promising properties such as cyto- and biocompatibility, antibacterial and antioxidant action, as well as increased curcumin solubility. Another system using bacterial cellulose to create hydrogels was bacterial cellulose hydrogels with embedded curcumin-loaded silver nanoparticles. The result of this study confirmed its antioxidant, wide-range antibacterial, and biocompatible properties. The wound healing properties of curcumin with the addition of silver nanoparticles in hydrogel systems were also analyzed in sodium alginate-co-acacia gum film hydrogels. The prepared complex was characterized by broad-spectrum antimicrobial activity and cell compatibility and created an appropriate environment for effective wound healing [47,48,49].

Curcumin can be grafted on various nanoparticles and further used to modify hydrogels. In one of the studies, a synthesized (through a one-pot synthesis technique) hydrogel system containing curcumin-loaded mesoporous polydopamine nanoparticles was exposed to NIR irradiation. It resulted in an active compound release. The analysis showed enhanced antimicrobial properties (against Gram-positive and Gram-negative bacteria; antibacterial effect reached 97.8% and 94.2% compared to control against *E. coli* and *S. aureus*, respectively), reduced inflammatory effect, and enhanced granulation of tissues and collagen deposition [50].

A curcumin nanoemulsion hydrogel-stabilized system was prepared by dispersing Carbopol 940 in water, followed by neutralization. The last step included stirring with curcumin nanoemulsion. The obtained system exhibited acceleration of wound healing in vivo in comparison to curcumin gel and commercially available nanosized curcumin gel. This effect may be associated with faster vascularization and epithelium recovery. It presented enhanced antibacterial action towards Gram-positive and Gram-negative bacteria (test against *E. coli* and *S. aureus*) and antioxidant activity [51].

Another study showed an oxidized cellulose nanofiber-poly(vinyl alcohol) hydrogel system, modified with curcumin micelles. The polymer system enhanced the curcumin absorption by L929 cells. The results of the following in vivo experiment showed improvement in the rate of wound healing without an increase in scar formation, due to the accelerated collagen organization rate (~80% of wound closure after 2 weeks for curcumin-containing hydrogels compared to ~40% for control and ~50% for hydrogel only) [52]. Curcumin liposomal formulation incorporated in lysine–collagen hydrogel also reported similar results. The combined action of lysine, collagen, and curcumin resulted in quicker healing without scar formation (complete healing after 7 days) accompanied by a noted anti-inflammatory effect (in vivo test on rats) [53]. Polymer-based hydrogels in a film form with incorporated curcumin are promising systems in wound treatment. Carrageenan and alginate hydrogel film functionalized with poloxamer curcumin was tested in vitro and in vivo and exhibited improved bioavailability of curcumin, a high safety profile, and enhanced cell proliferation properties which directly resulted in better wound healing (increased regeneration and reduced inflammation after 7 days of treatment with a curcumin-modified hydrogel compared the non-modified system) [45].

Using curcumin in hydrogel dressings instead of various nanoparticles (e.g., Cu, Ag) reduces the potential toxicity while maintaining the antimicrobial effect towards Gram-positive and Gram-negative bacteria. Other advantages of these materials include antioxidant properties, hemostatic effect, and enhanced collagen formation, which results in faster and better wound healing, including infected ones, which was confirmed in the in vitro and in vivo studies [46,47,54,55,56,57]. 

Diabetic wounds are very difficult to treat due to their susceptibility to infections—the glucose levels in the wound site are high, and the wound contains more moisture [58,59]. An in vivo study showed that curcumin nanoparticle hydrogels and curcumin hydrogels are effective in diabetic wound healing, but the nanoparticle version exhibits better results than curcumin hydrogels. It sped up the healing process, influenced collagen deposition and tissue formation more, as well as presented a more significant increase in VEGF (“vascular endothelial growths factor”) and AQP3 (“aquaporin-3”) [59]. Hybrid membrane camouflaged 4OI (4-octyl itaconate) nanovesicles in an injectable hydrogel system were synthesized using the extrusion method and analyzed for their diabetic wound healing properties. The results suggest their anti-inflammatory effect, enhancing vascularization properties, faster wound closure, and other important factors contributing to wound repair, such as improved epidermis and dermis regeneration or better collagen deposition [58]. The sustained curcumin release profile from the composites contributes to enhanced antibacterial properties, reducing the probability of infection and the healing time [53,60,61,62]. In vivo research showed that when synthesized using a hot, high-pressure homogenization technique, curcumin solid lipid nanoparticle hydrogels remain stable under autoclave conditions as well as photostable. It is safe and characterized by a prolonged, controlled release. Its improved antibacterial properties are associated with the *S. aureus* biofilm interference (inhibition of the biofilm formation at 512 µg/mL and against mature biofilm at 2048 µg/mL). It has anti-inflammatory and antioxidant activity and improved angiogenesis and granulation tissue formation, which speed up wound healing (complete wound closure after 11 days for curcumin solid lipid nanoparticles) [55].

The research concerning curcumin-modified hydrogels as a wound healing tool includes injectable formulations as well [63,64]. The pluronic-containing hydrogel modified with gelatin and curcumin is a biocompatible and biodegradable material that boosts the wound healing process which was proven in both in vitro and in vivo studies in different wound types, including second- and third-degree burns. Those injectable hydrogels are characterized by good physicochemical (e.g., rheological stability, drug release rate, modifiable gelation time) and biological (e.g., good blood clotting capacity, inflammation reduction, better collagen deposition, higher fibroblasts density, and granulation tissue thickness) properties [63,64,65,66]. All the analyzed studies show high biocompatibility and lack of cytotoxicity for all the investigated materials, proving their high safety profile [11,46,50,52,54,55,63,64,67].

### 2.3. Skin Diseases

Curcumin-modified hydrogels exhibit promising effects not only in wound treatment but can also be helpful against various skin diseases. One is dermatitis—skin inflammation that can occur in the acute or chronic form [68]. Curcumin’s anti-inflammatory properties and proven effect in wound healing make this compound a great candidate for dermatitis treatment. One study used a thin membrane hydration method to encapsulate curcumin into micelles. This process increased the compound’s solubility to 1.87 mg/mL, over 10^6^ times more than pure curcumin. Further preparation of the curcumin-loaded hydrogel resulted in enhanced skin permeation and skin deposition—after 12 h, curcumin hydrogel deposition was 6.23 times higher than the native compound. Compared with curcumin and dexamethasone (traditional treatment), curcumin hydrogel reduced inflammation by 58.6% more than curcumin and by 19.1% more than dexamethasone [68]. 

Psoriasis is an autoimmune, inflammatory disease with a complex etiology that impacts keratinocyte proliferation, differentiation, and dermal angiogenesis. It results in red, scaly lesions [69,70]. Standard treatment methods include phototherapy, systemic therapy, and topical therapy [71,72]. In a study, curcumin was initially entrapped in macrocycle micelle, followed by hydrogel formation. Choline-calix [4]arene-based curcumin nanohydrogel was investigated in vivo in mice. The results indicated that use of a curcumin-containing hydrogel inhibited most of the histopathological changes induced by imiquimod. The non-treated group, exposed to imiquimod, exhibited significant damage: hyperkeratosis, parakeratosis, acanthosis, and epidermal infiltrates. Another indicator was expression of zo-1 and occludin, which contribute to cell permeability. Those markers were significantly decreased in the imiquimod group, but in the curcumin hydrogel group, their levels were restored to nearly physiological levels. The curcumin hydrogel group reduced the amount of mast cells and their degranulation compared to control. TNF-α and il-1β, inflammatory cytokines were strongly elevated in the control group and moderated in the curcumin-treated group. The curcumin hydrogel group also exhibited antioxidant effect by inhibiting iNOS levels (Table 2) [72]. 

In another study, the preparation of curcumin-loaded nano-hydrogel included adding water to curcumin dissolved in oil and SMIX. Similarly, the prepared hydrogel system exhibited potential in the topical treatment, as its efficacy in treating psoriasis was better than the corticosteroid anti-psoriatic drug (improvement after 4 days of treatment) group [73]. Curcumin encapsulation in epichlorohydrin-β-cyclodextrin and hydrogel system formation resulted in improved curcumin activity, such as skin penetration and release pattern. Moreover, the conjugate exhibited anti-inflammatory properties (IL-6 in HaCaT cells increased compared to the control) [74]. The desirable properties, such as deeper penetration, better skin hydration, curcumin dispersion, and stable release were also reported when the hydrogel system contained nanoparticle-entrapped curcumin, confirmed by in vitro and in vivo studies. This system was synthesized in two stages. The first resulted in creating curcumin-loaded nanoparticles (preparation of which included two methods: anti-solvent and flash precipitation technique). During the second one, hydrogel was created [75].

### 2.4. Cancer

Cancer is one of the two leading causes of death in the world [76]. There are many types of cancer, and the most frequent types include breast, lung, and prostate cancer, while mortality rate is highest for lung, liver, stomach, and breast cancer [77]. Currently, there are many approaches to cancer treatment, including surgery, radiation, chemotherapy, or their combination [78,79]. Unfortunately, the therapeutic outcomes are often non-satisfactory, and recurrence of the neoplastic changes is frequent. Furthermore, existing chemotherapeutic treatments cause dangerous and devastating side effects and may result in drug-resistance development [78]. Using compounds derived from plants is a promising approach to create novel anticancer therapeutics [79]. Curcumin was proven to exhibit activity against various types of cancer—it influences carcinogenesis and apoptosis while being safe and non-toxic [2,8,79,80,81]. Again, the factors hampering its broader use are its unfavorable physico-chemical properties. Many formulations were developed to overcome this issue, amongst which hydrogels clearly stand out.

There are many approaches to curcumin–hydrogel formulation in cancer treatment. One of the most researched forms can be classified as injectable hydrogels, which incorporate various types of curcumin modification [82,83,84,85,86]. Glioma is an aggressive form of brain tumor. Due to its location, the treatment is complicated, and therapeutic options are limited. In one of the studies, a curcumin mPEG-PLA (polyethylene glycol-b-polylactide) nanopolymerosome-loaded PCLA-PEG-PCLA (poly(ε-caprolactone-co-lactide)-b-poly(ethylene glycol)-b34 poly(ε-caprolactone-co-lactide)) injectable hydrogel was prepared and implemented into the xenograft tumors in nude mice. The results demonstrated increased anti-tumor activity compared to control groups [82]. An injectable hydrogel composite was also tested against colorectal cancer. Anticancer activity was enhanced by entrapping hydrophobic curcumin in the Pluronic F127 micelles and incorporating hydrophilic 5-fluorouracil in the hydrogel network creating dual-drug system. Results of the study indicated a synergistic effect—the novel composite exhibited better efficacy against HT-29 cells (Figure 5) [83]. 

Another injectable photo/thermosensitive hydrogel was tested against malignant bone cancer—osteosarcoma. Hybrid material, consisting of curcumin-loaded microsphere hydrogel (Cur-MP/IR820 hybrid hydrogel), was successful in osteosarcoma treatment (tumor cells viability was the lowest-23.9% while using Cur-MP/IR820 gel with laser irradiation, while in the controls the viability percentage was higher—MC gel (115.1%), Cur-MPs gel (96.0%), and IR820 gel with laser irradiation (55.8%)). In the animal model, the hydrogel was injected around the tumor site and exposed to 808 nm laser irradiation. In the Cur-MP/IR820 + laser irradiated group, tumors were strongly ablated and eradicated with bone tissue repaired. Thanks to the thermo-sensitive hydrogel form, curcumin can be injected and delivered to the desired site, and localized drug concentration is guaranteed for a long time [84]. The exact property was used when preparing an injectable curcumin-thermosensitive hydrogel. It was synthesized by mixing poloxamer buffer solution with PEG400 and curcumin 1,2-propanediol solution and tested in vitro and in vivo in a liver cancer model. The results confirmed good physicochemical properties, and efficiency in treating cancer (54.87% inhibition rate) [85]. There are several types of melanoma; among them, uveal melanoma is one of the most frequent intraocular tumors. Early intervention and stable drug delivery is a key factor in therapy. The injectable hydrogel was created by incorporating curcumin-loaded polymeric nanoparticles in the collagen II and HA hydrogel. Results of the research showed the sustained release of the drug, even biodistribution, and good efficacy against uveal melanoma (45% MP-38 cells viability after 3 days). Moreover, the safety profile of the composite and its biocompatibility were determined to be high [86].

Another type of skin cancer, melanoma, is characterized by a high mortality rate. Transdermal delivery systems are promising formulations due to ease of administration and localized drug delivery. In a study, curcumin was entrapped in the hydroxypropyl-β-cyclodextrins. In the second step, those inclusion complexes were incorporated in the hydrogels. The research outcome showed enhanced photostability, better solubility, and transdermal permeability. Additionally, cytotoxicity studies confirmed the efficacy against melanoma cells (IC_50_ = 29 µg/mL) [87]. Squamous cell carcinoma also belongs to the skin cancer group. Graphene oxide is characterized by unique properties, such as a high surface-area-to-volume ratio or ability to form hydrogen bonds and electrostatic interactions, which makes it a promising compound in drug delivery. Creating curcumin and graphene oxide incorporated in alginate hydrogels in a film form resulted in a prolonged drug release profile and strong anticancer and cytotoxic activity (curcumin incorporation decreased the number of live cells compared to the control). The graphene oxide component of the composite provided enhanced stability, while curcumin improved the biocompatibility [88]. In one of the in vitro studies, curcumin-loaded PVA/cellulose nanocrystals were incorporated into a hydrogel and tested for potential use in breast and liver cancer lesions. Results confirmed the selective anticancer properties showing cytotoxicity against cancer cell lines (Huh-7, MCF-7) while being relatively non-toxic to human fibroblast cell lines (HFB-4). Moreover, the bioavailability of curcumin was improved due to the stable release profile from the membrane. Furthermore, the composite exhibited promising antimicrobial activity (with the most significant inhibition effect on *C*. *albicans* (31 mm), *K. pneumoniae* (29 mm), *E. aerogenes* (28 mm), and *B*. *cereus* (26 mm)) [79].

A curcumin-poly(ethylene oxide) hydrogel was prepared through polymerization of curcumin, PEG, and DTE catalyzed by triphosgene and pyridine and was investigated as a bioactive soft-tissue filler. The results of the study indicated good pharmacological properties and anti-tumor activity. This study shows that hydrogels may render curcumin available not only for treatment but—due to the high concentration in the application spot—also for the prevention and as an adjuvant anticancer therapy to prevent the relapse of neoplastic changes [89]. It is worth noting the time over which curcumin was released from this hydrogel, which was 80 days included in the study. This enormously exceeds the time that curcumin can exert action in the organism, which may be as short as 30 min for curcumin itself [90].

Curcumin hydrogels have also been tested for conventional use. One of potential applications is osteosarcoma treatment, which involves curcumin-loaded chitosan nanoparticles hyaluronic acid/silk fibroin hydrogel (CCNPs-SF/HAMA hydrogel). Firstly, the chitosan nanoparticles were synthesized followed by hydrogel formation. Results from an in vitro study confirmed the anti-osteosarcoma activity (cell viability decreased from 125% to 87% when curcumin concentration increased from 150 to 400 µg/mL), influence on bone proliferation, and suitable physical properties [91]. Another research study tested the anticancer effect on A549 lung adenocarcinoma cells. Using an electrostatic field, the curcumin was incorporated into three different biopolymer nanoparticles (chitosan, gelatin, hyaluronan). All the analyzed hydrogels exhibited higher efficacy on A549 cells than curcumin alone [92]. Hepatocellular carcinoma is the second most common type of cancer leading to death. There are few effective treatment protocols against this disease. Glycyrrhetinic acid is a natural triterpenoid that has been proven to bind specifically to liver cell membrane. This feature was utilized in a study where glycyrrhetinic acid-modified curcumin was incorporated into a hydrogel using a pro-gelator, thus forming a delivery system targeted to liver cancer cells. The obtained system was tested in vitro, and results showed that it increased cellular uptake by HepG2 cells and had better anti-cancer properties than the control [93].

Ionic liquids exhibit a remarkable ability to dissolve polymers. This fact was used to prepare curcumin-loaded polysaccharide-based hydrogel complexes. Firstly, the ionic liquid ([HMim][HSO4^−^]) was synthesized, and chitosan and curcumin were dissolved separately with chondroitin sulfate. Finally, both solutions were mixed to yield the novel material. Thanks to using an ionic liquid as a solvent, the obtained concentrations of polymer were higher compared to the water-dissolved ones and the hydrogel carrier ensured controlled release of the compound. The biological evaluation studies showed its safety towards healthy cell lines and cytotoxic effect against cancer cells (HeLa, HT29 and PC3) [38].

Hydrogels allow to enclose more than one compound in their matrix. Frequently, in hydrogel-delivery systems, curcumin is co-administered to maximize the therapeutic efficacy of the hydrogel. One such research system was based on the co-delivery of curcumin and doxorubicin—a clinically approved anticancer drug [94]. The designed hydrogel consisted of chitosan, amino-appended graphene, and amino-appended cellulose nanowhisker, all of which were crosslinked with a synthesized dialdehyde, 1,1’-(propane-1,3-diyl)bis(4-formylpyridin-1-ium) dibromide. The prepared drug delivery system was capable of releasing the carried substances dependent on the pH, promoting the release in acidic conditions (pH = 5.4).

The self-assembly systems can self-organize to form stable structures, such as micelles, solid lipid nanoparticles, and hydrogels, in response to external factors (temperature, pH, or contact with some solvents) [95,96]. To take advantage of those properties, the injectable curcumin-MAX8 peptide hydrogel system, capable of self-assembling under physiological conditions, was developed. This innovative drug delivery system was tested in vitro, and the results confirmed its efficacy against cancer cell lines (caspase 3 and PARP dose-dependent cleavage) and minimal level of invasiveness due to the localized distribution of the drug [40]. The incorporation of poly(lactic-co-glycolic acid) (PLGA) pellets into thermo-responsive hydrogel was conducted using two techniques: solvent evaporation and extrusion spheronization. It resulted in a long-term and stable drug release without significant burst release after administration [97]. The injectable hydrogels can solidify in situ and are characterized by low toxicity and high loading capacity. Encapsulating curcumin in β-cyclodextrin/ethylene glycol with addition of Pluronic 127 using the cold method resulted in the creation of thermoresponsive injectable hydrogel with good physicochemical properties and toxicity against cervical adenocarcinoma (HeLa) and mammary gland adenocarcinoma (MCF-7) cancer cells [98]. A similar material, curcumin-containing Pluronic 127-based thermosensitive hydrogel (PF127-g-AMPS), was synthesized by combining two methods, cold and free radical polymerization, and presented enhanced bioavailability, stability, and cytotoxicity against cancer cells [99]. 

### 2.5. Alzheimer’s Disease

Alzheimer’s disease is a chronic, neurodegenerative, and inflammatory disease that leads to amyloid β-proteins deposition in the brain, tau protein hyperphosphorylation, and neuronal damage resulting in impaired cognition. The multilateral activity of curcumin also includes anti-amyloid, antioxidant, and neuroprotective action. These properties seem to have great potential in Alzheimer’s disease therapy, but again, the low bioavailability and instability hinder their effectiveness [100,101,102]. There are several studies concerning various curcumin–hydrogel formulations for Alzheimer’s disease therapy. Among them are self-assembled hydrogels containing curcumin–hyaluronic acid conjugates. The combination of hydrophobic curcumin and hydrophilic hyaluronic acid increased curcumin’s inhibitory effect on amyloid formation and enhanced its neuroprotective activity [100]. Thermo-responsive hydrogels were also researched to confirm their potential against Alzheimer’s disease in in vitro and in vivo [101,102]. Examples of those hydrogels are curcumin-loaded mesoporous silica nanoparticles incorporated into a hydrogel and curcumin-loaded PLGA-PEG-PLGA hydrogel [101,102]. In one of the studies, to create a thermo-responsive mucoadhesive nasal delivery system, curcumin was encapsulated into mesoporous silica nanoparticles, which were in turn incorporated into a hydrogel matrix. In vitro and in vivo studies confirmed the functionalized hydrogel mucoadhesive properties and bioavailability. The test on streptozotocin-induced Alzheimer’s disease mice model confirmed the improvement in cognitive functions [101]. Another study concerned an injectable curcumin-loaded PLGA-PEG-PLGA hydrogel (Figure 6). Results of the research presented positive outcomes in reducing oxidative stress, amyloid β fibrillation and their cytotoxicity in vitro, as well as preventing neurodegeneration in vivo [102].

### 2.6. Protective and Regenerative Properties

Throughout their lives, people are exposed to many toxic factors. There are many sources of toxins such as food or drugs. One of them is aflatoxin B_1_—a widely occurring mycotoxin which can be found in cereals and grains. It is classified as a group one carcinogen due to its genotoxic, immunosuppressive, cytotoxic effect, and contribution to hepatocellular carcinoma formation. Creating curcumin nanoparticle-loaded hydrogels using a three-step process resulted in obtaining orally administered composites with good solubility and bioavailability properties. Research revealed a high safety profile and protective activity by contraindicating damage caused by aflatoxin B_1_ such as elevated DNA fragmentation in the liver and spleen, histopathological modifications in the liver, alterations in the bone marrow, or biochemical changes [103]. 

Another study tested a curcumin-loaded magnetic poly(*N*-isopropylacrylamide-co-methacrylic acid) hydrogel for its cardioprotective action in vitro and in vivo. The preparation of this system included hydrogel dissolution in NaOH with the addition of magnetic nanoparticles and curcumin followed by the addition of freeze-dried nanogel. The study showed decreased heart failure and cardiac hypertrophy markers (ANP, β-MHC, BNP) levels. GPX and SOD markers were also downgraded, which confirmed the anti-inflammatory and antioxidant properties of the analyzed system [104]. Another in vitro study investigated an injectable nanocurcumin/arginine chitosan hydrogel for hypoxia-induced endothelial dysfunction. The hydrogel was obtained by sequentially adding nanocurcumin and later arginine to the chitosan hydrogel, which had previously been neutralized with NaOH solution. Curcumin was used for its strong antioxidant ability, L-arginine for its eNOS-related activity, and chitosan for its rheological features. The obtained system exhibited a higher antioxidant effect than the control group, a sustained release profile in the acidic environment, and good cytocompatibility. The novel hydrogel enhanced nitric oxide release in the research, suppressing oxidative stress-induced endothelial dysfunction, preventing tube width reduction. The phosphorylated eNOS expression in the hypoxic environment was alleviated to a normal level after the treatment of the nanocurcumin/arginine chitosan hydrogel. The composite exhibited satisfying results, which suggest its capability to reverse hypoxia-induced damage on endothelial cells under an ischemic site [105]. 

Curcumin-loaded hydrogels were also evaluated for their regenerative properties. Retinal pigment epithelium (RPE) is a part of the eye that contributes to maintaining its structure. Curcumin/alginate hydrogels with incorporated RPE mice-derived cells were examined in vitro for their use in RPE regeneration. The hydrogel’s structure exhibited biocompatibility and favorable cell growth and adhesion properties. The evaluated system showed the ability to regulate the viability of the cells and enhance extracellular matrix formation by gene expression [106]. 

Intervertebral disc degenerative disease is a musculoskeletal disorder resulting in the degeneration of intervertebral discs with no effective treatments available. Gelatin methacrylate nanocurcumin-loaded hydrogels were evaluated for their regenerative properties. They were prepared using a solvent emulsion diffusion low-temperature solidification technique. The study showed a decrease in NF-κB expression, which resulted in anti-inflammatory and anti-apoptotic action. Moreover, the complex enhanced Collagen type II restoration and aggrecan expression in rats. Results of the research suggest its potential for further investigation in intervertebral disc degenerative disease treatment [107]. Another study confirmed the effectiveness of injectable curcumin-loaded poly(ethylene oxide) dimethacrylate-gelatin methacrylate hydrogel microgels, synthesized using a micro-fluidic device, on cartilage repair in osteoarthritis. In vitro studies indicated the attenuation of inflammatory markers and stem cells growth enhancement. Additionally, in vivo results showed cartilage regeneration [108]. This group of composites also exhibits promising effects on bone regeneration. The curcumin and bone morphogenetic protein-2 containing hyaluronic acid/poly-L-lysine hydrogels were synthesized and evaluated for their osteogenic properties with positive results, involving good proliferation properties and a sustained release profile. Moreover, analyzed hydrogels had a positive influence on early (alkaline phosphatase activity) and late (calcium deposition) markers of osteoblasts deposition as well as bone regeneration confirmed by micro-CT test, which suggests its suitability for future use as biomaterials in bone regeneration [109].

It is very difficult to treat traumatic spinal cord injury (SCI) due to its complexity caused by extensive acute inflammation. A multifunctional hydrogel was synthesized to create an effective treatment protocol. The hybrid curcumin-containing injectable, self-healing hydrogel exhibited good physicochemical properties. An in vitro study sped up DRG (“dorsal root ganglia”) neurite regeneration and Schwann cell (SC) migration. A recently developed hydrogel facilitated the local reassembly of the extracellular matrix, leading to a reduction in inflammation in vivo. Furthermore, the synthesized hydrogel promoted the recruitment of endogenous Schwann cells (SC) into its structure, thereby influencing their contribution to the process of newly regenerated nerve remyelination [110].

Curcumin presents a positive impact on gut microbiota. However, direct ingestion seems of little importance due to its instability. The effect is more pronounced when co-administered with probiotics, as synergy in action is observed to influence digestive health positively. Knowing this, Su et al. incorporated both curcumin and probiotics (*Lactobacillus rhamnosus* GG (LGG)) into a hydrogel in two steps. First, diluted β-lactoglobulin nanoparticle suspension was mixed with propylene glycol alginate solution. Next, the hydrogel formation was promoted after adding an ethanolic solution of curcumin. The results confirmed higher viability of LGG, slower release of curcumin in gastric fluid, increased release in the intestinal juice, and improved curcumin stability [29].

A series of hydrogels acting in a pH-dependent manner was researched. As curcumin is stable only in neutral and acidic environments and undergoes rapid degradation in an alkaline medium, the possibility of releasing the carried polyphenol only in the desired target site may significantly improve the biological effect exerted by curcumin. It was found that the pH level significantly modifies the drug release in zinc(II) oxide/curcumin hydrogel systems. Zinc(II) oxide is broadly studied in its nanoparticular form to provide a drug delivery vehicle [111,112]. Numerous studies have involved ZnO in curcumin-incorporating hydrogel systems. Such a combination improved curcumin stability and bioavailability and influenced the drug’s release time depending on pH level [111,112]. Hydrogel beads containing omeprazole and curcumin further coated with chitosan were obtained using a modified solvent kneading method and were found to be a pH-dependent drug delivery system as well. The synergistic effects of those two compounds resulted in better efficacy for peptic ulcers treatment than the traditional therapy, and the hydrogel formulation contributed to the most effective drug release at pH = 1.2, which is close to the pH level within the stomach [113]. Another chitosan-based hydrogel was functionalized to develop a carboxymethyl chitosan/Fe_3_O_4_ nanocomposite containing curcumin. This magnetic system presented a positive outcome in drug release in a pH dependent manner [114]. In situ pH-responsive hydrogels incorporated into polyhydric solvents exhibit great potential for gastric diseases such as cancer and ulcers. Those polyhydrogels use the cosolvency effect, in which the properties of the gel matrix undergo an alteration due to the interactions with the appropriate solvent. This phenomenon enables the prepared composite to incorporate hydrophobic drugs that show activity against stomach diseases [115]. Poly(sucrose acrylate-co-polymethylacrylic acid) hydrogel oral delivery system was developed using free radical polymerization and evaluated for its properties. Research revealed its pH-dependent release profile. Conversely to the majority of the studies described so far, curcumin was released in an alkaline environment, making it a valuable system for curcumin delivery in the lower parts of the gastrointestinal tract [116].

## 3. Conclusions and Perspectives

Curcumin is a very promising natural compound with a high safety profile. Unfortunately, its therapeutic potential is limited by poor ADME properties. Creating curcumin–hydrogel formulations significantly improves the bioavailability of curcumin and its use for pharmaceutical purposes. Such a combination exhibits better activity in in vitro and in vivo studies than curcumin by itself. This is due to a couple of associated effects, mainly better curcumin solubility in these formulations and improved curcumin stability resulting from the hydrogels’ protective environment. Other than that, the hydrogel matrix allows the embedding of more than one compound, enhancing the pharmacological effect of the obtained formulations.

Developed systems have great potential in medicinal applications using various administration modes, including oral, injectable, intranasal, or transdermal. Additionally, the obtained formulation can present a desired release profile depending on the preparation method. Additionally, hydrogels can be modified to release curcumin in response to external factors that can be used in the targeted therapies.

While the number of conducted studies is high in the cancer field, there is still room for development in other areas, especially in neurodegenerative and inflammatory diseases. The curcumin–hydrogel research should be expanded not only in the current fields of interest but also in the new treatments by taking into the consideration compound’s versatile biological properties, the hydrogel’s unique physicochemical properties, and the possibility to incorporate other additional compounds in hydrogel formulations.

On the other hand, a big part of the studies dealing with such systems focus strictly on laboratory tests to evaluate the formulations using models or drug release in simulated conditions, mainly from the point of view of materials science. Further assessment using at least cell lines in vitro is necessary to provide an insight into the safety profile of those systems. At the same time, the pharmacokinetic parameters are difficult to predict when not conducted using animal models. Also, when it comes to the hydrogels of synthetic origin, little is reported on their fate post-therapy, whether they are eliminated from the body, and how they affect the organism upon chronic exposure. As for the prolonged action, due to the rapid elimination of curcumin encountered so far, it is difficult to predict what the effects of high doses of highly bioavailable polyphenols will be on healthy organisms. They may be difficult to predict, as some free radicals also participate in the physiological pathways and signaling in the human organism (i.e., NO). It is also worth mentioning the curcumin’s capability to chelate ions, which might lead to a shortage of cofactors for proper enzyme action.

Other drawbacks of hydrogels might include the unpredicted leaking of cross-linking agents, which usually contain reactive dialdehydes, which are generally toxic for humans. Also, regarding synthetic hydrogels, one must consider the environmental fate of such systems. They are more resistant to external factors, and, thus, might pose a threat to the natural environment, for example, due to accumulation in aquatic organisms. Still, little is known on this subject, and the upcoming research should consider such a phenomenon.

All of that suggests that the coming years will bring new, presumably multiple modifications of curcumin-bearing hydrogels, and more effort to be put into the biological assessment of such an approach.

## Figures and Tables

**Figure 1 molecules-29-01757-f001:**
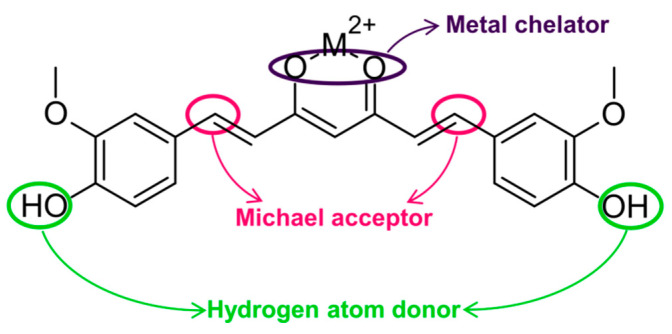
Curcumin chemical structure and its reactive sites.

**Figure 2 molecules-29-01757-f002:**
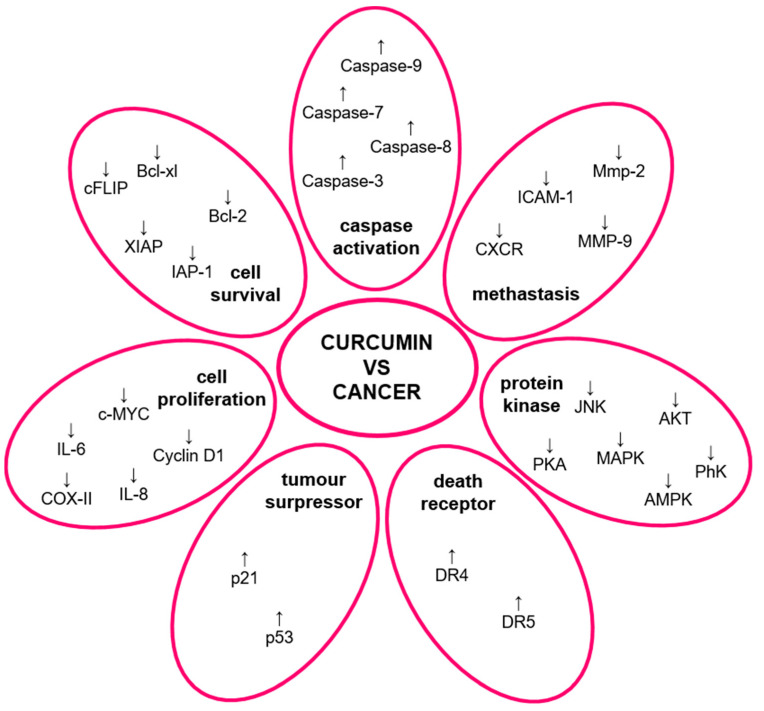
Influence of curcumin on molecular pathways related to cancer.

**Figure 3 molecules-29-01757-f003:**
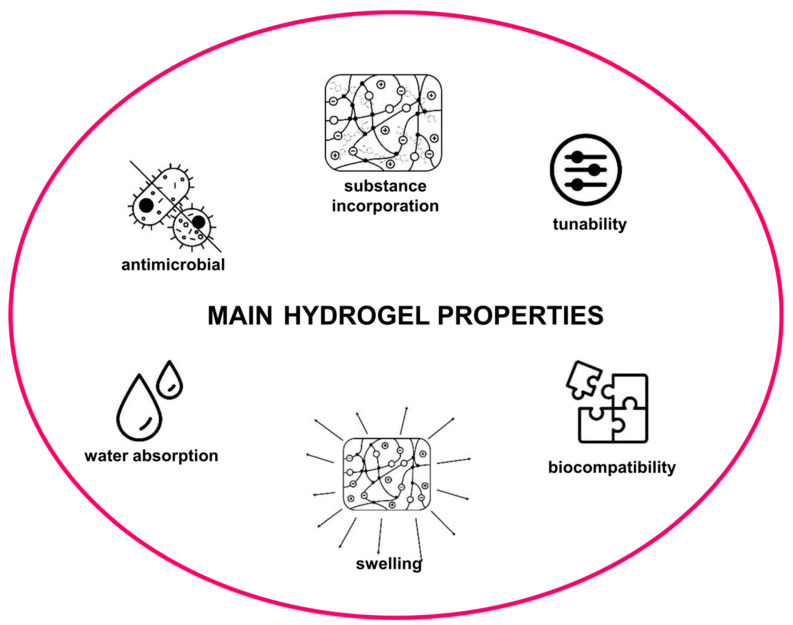
Main hydrogel properties.

**Figure 4 molecules-29-01757-f004:**
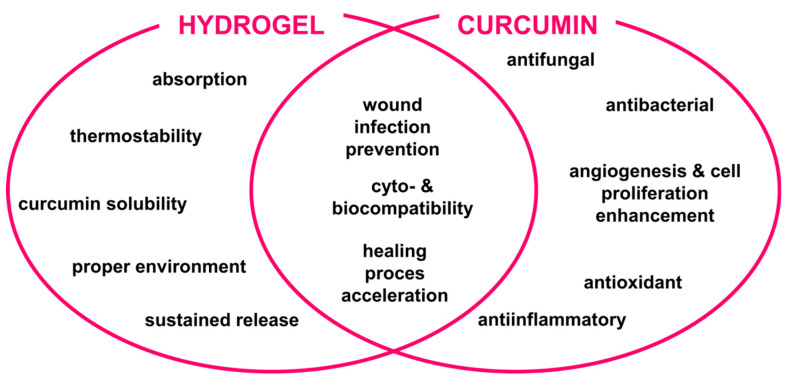
Effects exerted by curcumin and hydrogel that could contribute to the improvement in wound dressing.

**Figure 5 molecules-29-01757-f005:**
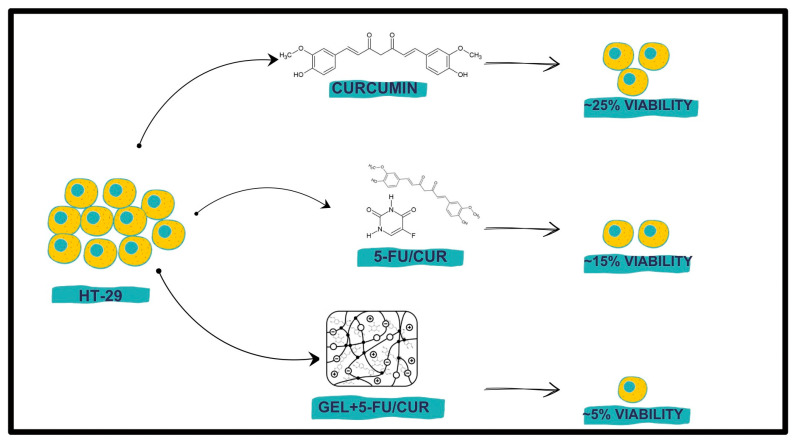
Anticancer effect of 5-fluorouracil/curcumin hydrogel system [83].

**Figure 6 molecules-29-01757-f006:**
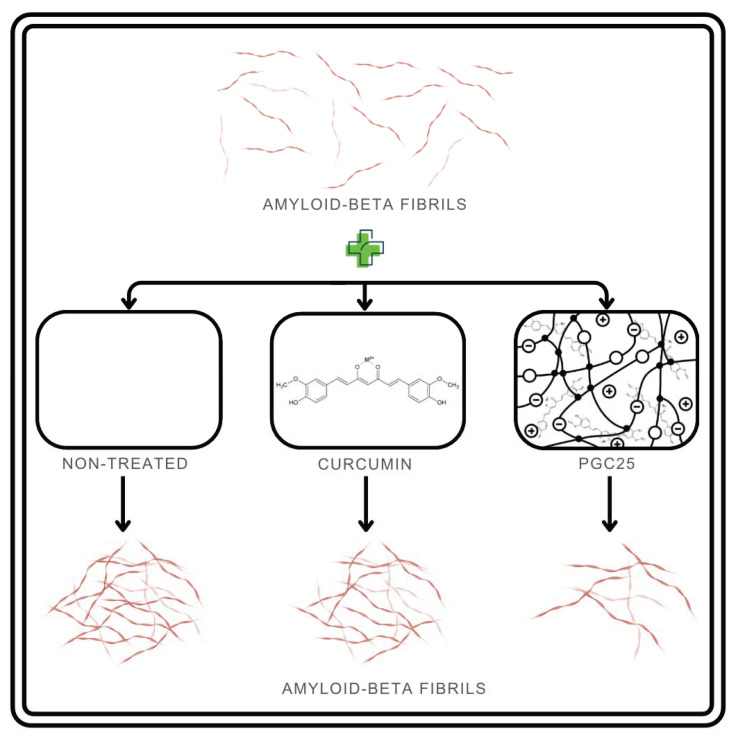
Inhibition of amyloid fibril aggregation by curcumin-containing hydrogel as reported in [102].

**Table 1 molecules-29-01757-t001:** Selected forms of curcumin–hydrogel based on the administration route and main improvements they offer.

Type of Curcumin–Hydrogel	Main Advantage	References
Nano-based oral delivery systems	Controlled/prolonged release	[30,32]
Nano-based topical delivery systems	Better skin permeability	[31,33]
Injectable delivery systems	Controlled release	[34,35]

**Table 2 molecules-29-01757-t002:** Effects of psoriasis treatment with the calix [4]arene-based curcumin nanohydrogel [72]. Imiquimod is an inducer of psoriasis-like changes in BALB/c mice.

	Histo- Logical Score	ZO-1 Expression	Occludin Expression	Mast Cell Proliferation	TNF-α	IL-1β	iNOS Levels
		[% of Total Tissue Count]		[Number/mm^2^]	[% of Total Tissue Count]		
Control	ND	~7%	~7%	~0	~0%	~0%	~0%
Imiquimod	~4	~2%	~1.5%	~50	~7%	~7.5%	~7%
Imiquimod + hydrogel	~1	~5%	~5%	~10	~2%	~2%	~2%

## Data Availability

Data are contained within the article.

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
