# Peer review of "Wondrous Yellow Molecule: Are Hydrogels a Successful Strategy to Overcome the Limitations of Curcumin?"

_molecules, 2024, doi:10.3390/molecules29081757_

Round 1

Reviewer 1 Report

Comments and Suggestions for Authors

This paper provides an overview of curcumin hydrogel formulations used in various applications such as wound healing, skin diseases, anti-cancer and regenerative activities. Hydrogel formulation improves curcumin biopharmaceutical properties.

In my opinion, the manuscript is well written, well organised and includes many examples. The bibliography is updated, with most references between 2018 and 2023.

I have a few comments that I hope will help the authors:

-          Abbreviations such as JECFA, EPSA (line 76), GG, LGG (line 185), 4OI (line 313), VEGF, AQP3 (line 321), PEG-PLA-PCLA (line 398), DRG (line 538) should be described the first time they are used.

-          Line 209, Fe3O4, it is better to use subscripts

-          In Table 1 I think it would be useful to give the relevant references next to each of the items.

-          Please revise the sentence referring to Doxorubicin (lines 491-492). I think that the reference to doxorubicin can be removed, there is nothing related to it in the manuscript, there is no study mentioned, not even on a comparative level with curcumin.

Comments on the Quality of English Language

  - Please revise the sentences in lines 267-269, 307-310, 331-334 (composite?).

-         In line 289 retype “characterization”, in line 335 “characterized” and in line 472 “thermo-responsive”.

Author Response

Response to Reviewer 1

1. Summary

Thank you very much for taking the time to review this manuscript. Please find the detailed responses below and the corresponding revisions in track changes in the re-submitted files.

2. Point-by-point response to Comments and Suggestions for Authors

This paper provides an overview of curcumin hydrogel formulations used in various applications such as wound healing, skin diseases, anti-cancer and regenerative activities. Hydrogel formulation improves curcumin biopharmaceutical properties.

In my opinion, the manuscript is well written, well organised and includes many examples. The bibliography is updated, with most references between 2018 and 2023.

I have a few comments that I hope will help the authors:

Comments 1:

Abbreviations such as JECFA, EPSA (line 76), GG, LGG (line 185), 4OI (line 313), VEGF, AQP3 (line 321), PEG-PLA-PCLA (line 398), DRG (line 538) should be described the first time they are used.

Response 1:

We are very grateful for the reviewer for the time spent on this manuscript. All the abbreviations that appear in-text have their explanation is now given.

Comments 2:

Line 209, Fe3O4, it is better to use subscripts

Response 2:

We are sorry for the typo. This and other errors were corrected throughout the text.

Comments 3:

In Table 1 I think it would be useful to give the relevant references next to each of the items.

Response 3:

A third column was added to the table with the corresponding references.

Comments 4:

Please revise the sentence referring to Doxorubicin (lines 491-492). I think that the reference to doxorubicin can be removed, there is nothing related to it in the manuscript, there is no study mentioned, not even on a comparative level with curcumin.

Response 4:

The mentioned fragment was entirely removed from the manuscript.

3. Response to Comments on the Quality of English Language

Point 1:

Please revise the sentences in lines 267-269, 307-310, 331-334 (composite?).

Response 1:    The sentence was corrected

Point 2:

In line 289 retype “characterization”, in line 335 “characterized” and in line 472 “thermo-responsive”.

Response 2:    The typos/errors were corrected.

5. Additional clarifications

Many typos and English errors were corrected throughout the text, including the mentioned.

Reviewer 2 Report

Comments and Suggestions for Authors

Thanks for editor

Comments about (Wondrous yellow molecule: are hydrogels successful strategy to improve the bioavailability of curcumin?)

1.       The Graphical Abstract (GA) has to be redesigned by the authors because it is unclear

2.       Concerning the study's uniqueness and justification,. I believe there has been a lot of research on curcumin in hydrogels and reviews. what is the innovation in the author's review?

3.       In the introduction, the authors mention more data, there is a repetition of data in the core of the reviewer

4.       Table 1 needs references for each mentioned data.

5.       It is better to mention the preparation,methods of different hydrogel   

Comments on the Quality of English Language

Thanks 

The review needs English editing to clarify some expressions mentioned by the authors.

Author Response

Response to Reviewer 2

1. Summary

Thank you very much for taking the time to review this manuscript. Please find the detailed responses below and the corresponding revisions in track changes in the re-submitted files.

2. Point-by-point response to Comments and Suggestions for Authors

Thanks for editor

Comments about (Wondrous yellow molecule: are hydrogels successful strategy to improve the bioavailability of curcumin?)

Comments 1:

The Graphical Abstract (GA) has to be redesigned by the authors because it is unclear

Response 1:

The graphical abstract was changed.

Comments 2:

Concerning the study's uniqueness and justification,. I believe there has been a lot of research on curcumin in hydrogels and reviews. what is the innovation in the author's review?

Response 2:

Based on the literature search, there is no such comprehensive review that would encompass the medical use of curcumin in hydrogel formulations (Scopus database, 25th March). The papers published during the last five years regarding similar topics do not actually cover the information presented in this review. If the main topic is curcumin, the papers are not focused entirely on hydrogels as formulations but on other drug delivery systems as well. Reviews focusing on hydrogels usually describe other naturally occurring compounds as carried substances, and curcumin is only partially covered by them. Finally, if the articles revolve around both hydrogels and curcumin, they are usually limited to anticancer activity and/or potential use in wound healing applications [1–6].

We aimed to focus the review on the combination of hydrogels and curcumin and the possible therapeutic uses for which such a combination could be applied.

Comments 3:

In the introduction, the authors mention more data, there is a repetition of data in the core of the reviewer

Response 3:

The introduction subchapter was modified, as well as the structure of the whole review in accordance with reviewers’ comments.

Comments 4:

Table 1 needs references for each mentioned data.

Response 4:

A third column was added to the table with the corresponding references.

Comments 5:

It is better to mention the preparation, methods of different hydrogel  

Response 5:

The preparation methods of the hydrogels mentioned in the review were added.

References:

1.         Aroraa, S.; Dhoke, V.; Moharir, K.; Yende, S.; Shah, S. Novel Drug Delivery System of Phytopharmaceuticals: A Review. Current Traditional Medicine 637549920000000000, 07, doi:10.2174/2215083807666210426121038.

2.         Sethiya, A.; Agarwal, D.K.; Agarwal, S. Current Trends in Drug Delivery System of Curcumin and Its Therapeutic Applications. Mini Rev Med Chem 2020, 20, 1190–1232, doi:10.2174/1389557520666200429103647.

3.         Prasathkumar, M.; Sadhasivam, S. Chitosan/Hyaluronic Acid/Alginate and an Assorted Polymers Loaded with Honey, Plant, and Marine Compounds for Progressive Wound Healing-Know-How. Int J Biol Macromol 2021, 186, 656–685, doi:10.1016/j.ijbiomac.2021.07.067.

4.         Madamsetty, V.S.; Vazifehdoost, M.; Alhashemi, S.H.; Davoudi, H.; Zarrabi, A.; Dehshahri, A.; Fekri, H.S.; Mohammadinejad, R.; Thakur, V.K. Next-Generation Hydrogels as Biomaterials for Biomedical Applications: Exploring the Role of Curcumin. ACS Omega 2023, 8, 8960–8976, doi:10.1021/acsomega.2c07062.

5.         Rahmanian, M.; Oroojalian, F.; Pishavar, E.; Kesharwani, P.; Sahebkar, A. Nanogels, Nanodiscs, Yeast Cells, and Metallo-Complexes-Based Curcumin Delivery for Therapeutic Applications. European Polymer Journal 2023, 196, 112215, doi:10.1016/j.eurpolymj.2023.112215.

6.         Sood, A.; Dev, A.; Das, S.S.; Kim, H.J.; Kumar, A.; Thakur, V.K.; Han, S.S. Curcumin-Loaded Alginate Hydrogels for Cancer Therapy and Wound Healing Applications: A Review. Int J Biol Macromol 2023, 232, 123283, doi:10.1016/j.ijbiomac.2023.123283.

3. Response to Comments on the Quality of English Language

Point 1:

The review needs English editing to clarify some expressions mentioned by the authors.

Response 1: Many typos and English errors were corrected throughout the text.

Reviewer 3 Report

Comments and Suggestions for Authors

Though the title refers to hydrogels to improve the bioavailability of curcumin, the review provides little information to analyze how the hydrogel improve the bioavailability. Dose material, preparation method, or drug amount affect the bioavailability? These points need to be discussed in detail.

Author Response

Response to Reviewer 3

1. Summary

Thank you very much for taking the time to review this manuscript. Please find the detailed responses below and the corresponding revisions in track changes in the re-submitted files.

2. Point-by-point response to Comments and Suggestions for Authors

Comments 1:

Though the title refers to hydrogels to improve the bioavailability of curcumin, the review provides little information to analyze how the hydrogel improve the bioavailability. Dose material, preparation method, or drug amount affect the bioavailability? These points need to be discussed in detail.

Response 1:

Thank You for the insightful comment. However, the current state of the art makes it impossible to draw such conclusions. For the time being, it is known that a proper hydrogel formulation improves the biological action of curcumin due to the improvement of its chemical environment (protection from degrading conditions, improving solubility, enabling targeted delivery). As hydrogels exhibit porosity, the drug amount is not as important as the surface of the hydrogel forming polymers may be tuned, the preparation method might be considered, but so far, there are too few studies on the same composition of hydrogels that were prepared differently, or such studies are not reported. The problem arises when actual bioavailability is considered, as we can only rely on indirect data (such as biological activity, stability of curcumin, amount of drug release). The problem in question is crucial to understanding hydrogel formulation of curcumin but, to our best knowledge, cannot be answered yet.

Reviewer 4 Report

Comments and Suggestions for Authors

RecommendationMajor Revision

Comments:

This manuscript introduced various curcumin hydrogels, reviewed the wound healing properties, effectiveness in treating skin diseases, anticancer activity, and protective regenerative activity of these curcumin hydrogels, and pointed out hydrogels enhanced the chemical stability, bioavailability and water solubility of curcumin. However, the structure of the manuscript was not clear enough, the content was scattered, and some points lacked supportive arguments. The main questions were listed as follows:

1.     The central argument of the manuscript is "Hydrogels are successful strategy to improve the bioavailability of curcumin." The part of introduction should clarify the central argument. Please add or simplify some descriptions as appropriate to highlight the improved therapeutic effect of using hydrogels to deliver curcumin.

1) Simplify the description of curcumin in the introduction may better highlight the key points.

2) Please describe in detail the advantages of using hydrogels to deliver curcumin.

2.     The structure of the manuscript was not clear and needed to be reorganized into a clear framework. The parts from 2 to 8 were not in strict logical parallel. For example, the different hydrogel types could be classified by the preparation method, administration route, and types of diseases which could be treated, and further categorized and presented under each classification. Similarly, the contents of Table 1 need to be modified in accordance with the structural framework of the manuscript.

3.     The influence of curcumin on molecular pathways related to cancer mentioned in  Figure 2 could not correspond exactly to those in the main text section. For example, the effect of curcumin on cancer metastasis was mentioned in Figure 2, but not in the manuscript. And the death receptor pathways (DR4, DR5) were mentioned in the manuscript but not shown in Figure 2. Please match the information in Figure 2 to the main text section.

4.     On page 5, line 154, it is mentioned that "The biggest challenge in this formulation is curcumin insolubility ". However, the following example elaborated on the improvement of skin permeability of curcumin by hydrogel could not prove the point very well. Please explain the correlation between insolubility and skin permeability of curcumin.

5.     On the line 170 - 177 on page 6, the authors only cited examples but not present the point that the example tried to prove. Please further elaborate on the point in this paragraph.

6.     On the line 190 - 196 on page 6, it is mentioned that the main advantage of the self-assembled system is targeted delivery, but no research examples were given, please add it.

7.     It may be more convincing to use specific experimental data and experimental figures when giving examples. Please add figures of the experimental data results related to the examples which was cited to support the points.

8.     References should be cited according to the central argument. For example, the statements on page 11, lines 437 to 440, were not related to the argument that “the use of hydrogels to deliver curcumin may improve its bioavailability for better cancer treatment." Please check the full text and revise.

9.     In the part of "9. Conclusions and perspectives", the authors summarized the current researches of curcumin hydrogel, but the challenges of curcumin hydrogel in the clinical application should be further discussed. Please add it.

10.  Please standardize "12hermos-responsive" on line 472 of page 12, and check the full text for spelling mistakes.

11.  "DRG" appeared for the first time on page 13, line 538, but the full English name was not given. Please check the full text and revise.

12.  It is recommended that the English in the manuscript needs to be modified.

Comments on the Quality of English Language

It is recommended that the English in the manuscript needs to be modified.

Author Response

Response to Reviewer 4

1. Summary

Thank you very much for taking the time to review this manuscript. Please find the detailed responses below and the corresponding revisions in track changes in the re-submitted files.

2. Point-by-point response to Comments and Suggestions for Authors

This manuscript introduced various curcumin hydrogels, reviewed the wound healing properties, effectiveness in treating skin diseases, anticancer activity, and protective regenerative activity of these curcumin hydrogels, and pointed out hydrogels enhanced the chemical stability, bioavailability and water solubility of curcumin. However, the structure of the manuscript was not clear enough, the content was scattered, and some points lacked supportive arguments. The main questions were listed as follows:

Comments 1:

The central argument of the manuscript is "Hydrogels are successful strategy to improve the bioavailability of curcumin." The part of introduction should clarify the central argument. Please add or simplify some descriptions as appropriate to highlight the improved therapeutic effect of using hydrogels to deliver curcumin.

1) Simplify the description of curcumin in the introduction may better highlight the key points.

2) Please describe in detail the advantages of using hydrogels to deliver curcumin.

Response 1:

Ad. 1)The description of curcumin was shortened in the introduction as much as possible – essential parts were left to introduce the readers into the subjects discussed later in the review.

Ad. 2) The advantages of using hydrogels were described in the subchapter 2 and 2.1.

Comments 2:

The structure of the manuscript was not clear and needed to be reorganized into a clear framework. The parts from 2 to 8 were not in strict logical parallel. For example, the different hydrogel types could be classified by the preparation method, administration route, and types of diseases which could be treated, and further categorized and presented under each classification. Similarly, the contents of Table 1 need to be modified in accordance with the structural framework of the manuscript.

Response 2:

The structure of the manuscript was modified. The order of the chapters was maintained but the chapters themselves were changed.

In this review we focused on the biological target of the tested hydrogel or the disease that they were tested towards. Where possible, additional categorization was applied but oftentimes no strict pattern could be identified.

Comments 3:

The influence of curcumin on molecular pathways related to cancer mentioned in  Figure 2 could not correspond exactly to those in the main text section. For example, the effect of curcumin on cancer metastasis was mentioned in Figure 2, but not in the manuscript. And the death receptor pathways (DR4, DR5) were mentioned in the manuscript but not shown in Figure 2. Please match the information in Figure 2 to the main text section.

Response 3:

Both the text and the Figure were modified to include matching information.

Comments 4:

On page 5, line 154, it is mentioned that "The biggest challenge in this formulation is curcumin insolubility ". However, the following example elaborated on the improvement of skin permeability of curcumin by hydrogel could not prove the point very well. Please explain the correlation between insolubility and skin permeability of curcumin.

Response 4:

The permeation through the skin relies mostly on the solubility of curcumin in the case of application of the compound by itself or in a solution. As there are no specific receptors enabling the active transport of curcumin to the viable skin cells, the penetration relies on Fick’s first law of diffusion. This was shown in a series of studies that reported increased permeability in vitro and ex vivo in animal skin models [1]. In the case of hydrogel formulations, not only solubility has to be taken into account, but other factors as well, including the properties of the vehicle. In the case of using a delivery system, curcumin penetration may rely either on the increased solubility of curcumin [2] or on the penetration of the API delivery system. In the latter case, curcumin does not have to be dissolved, as it might be released from the vehicle after the whole system is absorbed [3], especially if other additives are present that induce stratum corneum fluidization – i.e., skin penetration promoters [4].

The sentence in question was rephrased to better reflect on the content of the paragraph:

“The biggest challenge in topical delivery of curcumin is its insolubility [31,35].”

Comments 5:

On the line 170 - 177 on page 6, the authors only cited examples but not present the point that the example tried to prove. Please further elaborate on the point in this paragraph.

Response 5:

The paragraph was modified:

“Injectable hydrogels are a good way to deliver the drugs to specific locations - the deep and shallow areas. Depending on hydrogel matrix substrates,  injectable drug delivery systems can serve different functions and release curcumin in the desired way [34,38]. Thiolated curcumin-loaded chitosan/poly(ethylene oxide) diacrylate (TCS/PEGDA) injectable hydrogel was synthesized by mixing TCS/PEGDA with curcumin and evaluated both in vitro and in vivo. The drug release profile was found to be stable and controlled. Curcumin was released in a prolonged way, which contributed to higher efficiency. The system was reported to be non-toxic while exhibiting antitumor activity (tumor inhibition values were greater than 40%) [34]. Another example of hydrogel with controlled drug release in the injectable form is oxidized cyclodextrin-functionalized injectable gelatin hydrogels, which have suitable physical, chemical, and biological properties confirmed in the in vitro studies. Its significant bioadhesive function (2-4-fold higher than fibrin glue) shows great promise in constricting the area of carried substance effect [35].”

Comments 6:

On the line 190 - 196 on page 6, it is mentioned that the main advantage of the self-assembled system is targeted delivery, but no research examples were given, please add it.

Response 6:

The questioned fragment was modified to better reflect our intention. The self-assembly systems are designed to assemble in response to a specific factor and consecutively release the carried compound. In the case of the given example, curcumin was entrapped in the peptide-based hydrogel that could be freely distributed in the organism due to self-assembly, while the release was not connected to the disassembly of the system.

Comments 7:

It may be more convincing to use specific experimental data and experimental figures when giving examples. Please add figures of the experimental data results related to the examples which was cited to support the points.

Response 7:

We thank the reviewer for the suggestion. The experimental data was added, however we did not introduce additional figures, as, in our opinion, it would make the article too lengthy and blur out the goal of the review.

Comments 8:

References should be cited according to the central argument. For example, the statements on page 11, lines 437 to 440, were not related to the argument that “the use of hydrogels to deliver curcumin may improve its bioavailability for better cancer treatment." Please check the full text and revise.

Response 8:

The use of long-acting delayed-release hydrogel, as mentioned in this fragment, greatly increases the bioavailability of curcumin. As curcumin undergoes rapid metabolism within the body (even 30 minutes after application [5]). By using the mentioned polymeric hydrogel, the presence of curcumin can be prolonged for up to 80 days. In this context, bioavailability is increased by its delayed metabolism and excretion from the body. Regardless, this passage and several others have been altered to emphasize the authors' intent.

Comments 9:

In the part of "9. Conclusions and perspectives", the authors summarized the current researches of curcumin hydrogel, but the challenges of curcumin hydrogel in the clinical application should be further discussed. Please add it.

Response 9:

This subchapter was expanded by adding a following text fragment:

“As for the prolonged action, due to the rapid elimination of curcumin encountered so far, it is difficult to predict what the effects of high doses of highly bioavailable polyphenols will be on healthy organism. They may be difficult to predict, as some free radicals also participate in the physiological pathways and signalling in the human organism (i.e., NO). It is also worth mentioning the curcumin’s capability to chelate ions, which might lead to a shortage of cofactors for proper enzyme action.

Other drawbacks of hydrogels might include the unpredicted leaking of cross-linking agents, which usually contain reactive dialdehydes, which are generally toxic for humans. Also, regarding synthetic hydrogels, one must consider the environmental fate of such systems. They are more resistant to external factors, and, thus, might pose a threat to the natural environment, for example, due to accumulation in aquatic organisms. Still, little is known about this subject, and the upcoming research should consider such a phenomenon.”

Comments 10:

Please standardize "12hermos-responsive" on line 472 of page 12, and check the full text for spelling mistakes.

Response 10:

Many typos and English errors were corrected throughout the text, including the mentioned.

Comments 11:

"DRG" appeared for the first time on page 13, line 538, but the full English name was not given. Please check the full text and revise.

Response 11:

For all the abbreviations that appear in-text have their explanation is now given.

References:

1.         Pelikh, O.; Pinnapireddy, S.R.; Keck, C.M. Dermal Penetration Analysis of Curcumin in an Ex Vivo Porcine Ear Model Using Epifluorescence Microscopy and Digital Image Processing. Skin Pharmacol Physiol 2021, 34, 281–299, doi:10.1159/000514498.

2.         Eckert, R.W.; Wiemann, S.; Keck, C.M. Improved Dermal and Transdermal Delivery of Curcumin with SmartFilms and Nanocrystals. Molecules 2021, 26, 1633, doi:10.3390/molecules26061633.

3.         Nafisi, S.; Maibach, H.I. Chapter 3 - Skin Penetration of Nanoparticles. In Emerging Nanotechnologies in Immunology; Shegokar, R., Souto, E.B., Eds.; Micro and Nano Technologies; Elsevier: Boston, 2018; pp. 47–88 ISBN 978-0-323-40016-9.

4.         Yousef, S.A.; Mohammed, Y.H.; Namjoshi, S.; Grice, J.E.; Benson, H.A.E.; Sakran, W.; Roberts, M.S. Mechanistic Evaluation of Enhanced Curcumin Delivery through Human Skin In Vitro from Optimised Nanoemulsion Formulations Fabricated with Different Penetration Enhancers. Pharmaceutics 2019, 11, 639, doi:10.3390/pharmaceutics11120639.

5.         Devkota, H.P.; Adhikari-Devkota, A.; Bhandari, D.R. Chapter4.7 - Curcumin. In Antioxidants Effects in Health; Nabavi, S.M., Silva, A.S., Eds.; Elsevier, 2022; pp. 341–352 ISBN 978-0-12-819096-8.

3. Response to Comments on the Quality of English Language

Point 1:

It is recommended that the English in the manuscript needs to be modified.

Response 1: English in the whole manuscript was revised.

Round 2

Reviewer 3 Report

Comments and Suggestions for Authors

Bioavailability refers to the extent and rate at which the active moiety (drug or metabolite) enters systemic circulation, thereby accessing the site of action. As Table 1 shows, the content of this review is not only related to the availability improvement. Therefore, it is suggested to change the title of this review. 

Author Response

Response to Reviewer #3

Bioavailability refers to the extent and rate at which the active moiety (drug or metabolite) enters systemic circulation, thereby accessing the site of action. As Table 1 shows, the content of this review is not only related to the availability improvement. Therefore, it is suggested to change the title of this review.

The title of the review was changed according to the reviewer’s suggestion.

The new title is: “Wondrous yellow molecule: Are hydrogels a successful strategy to overcome the limitations of curcumin?”.

Reviewer 4 Report

Comments and Suggestions for Authors

Recommendation: Accept

The authors have made great efforts to address our comments and our questions have been mostly answered. Hence, we recommend this manuscript to be accepted by “Molecules”. However, some questions still existed.

1. For comment 7: We recommended that add figures of the experimental results related to the examples which was cited to support the points. While some preparation methods of curcumin hydrogel were supplemented, as shown in lines 648-652 on page 15. This is not related to the central argument "Hydrogels are a successful strategy to improve the bioavailability of curcumin." Please check the full text and revise it.

2. On the line 635 - 663 on page 15, the authors only cited examples of pH-dependent hydrogels but not present the point that the example tried to prove. Please further elaborate on the point in this paragraph.

Author Response

Response to Reviewer #4

Recommendation: Accept

The authors have made great efforts to address our comments and our questions have been mostly answered. Hence, we recommend this manuscript to be accepted by “Molecules”. However, some questions still existed.

  1. For comment 7: We recommended that add figures of the experimental results related to the examples which was cited to support the points. While some preparation methods of curcumin hydrogel were supplemented, as shown in lines 648-652 on page 15. This is not related to the central argument "Hydrogels are a successful strategy to improve the bioavailability of curcumin." Please check the full text and revise it.

The figures containing the preparation methods of hydrogel-embedded curcumin was not added, as in our opinion such change would not bring too much information regarding the central argument of the review, as well as it would make the manuscript too lengthy and blur the main findings of the study.

As for the description of the preparation methods in-text, the whole text was review and some of the preparations were removed.

  1. On the line 635 - 663 on page 15, the authors only cited examples of pH-dependent hydrogels but not present the point that the example tried to prove. Please further elaborate on the point in this paragraph.

An explanation was added to this paragraph: “As curcumin is stable only in neutral and acidic environments and undergoes rapid degradation in an alkaline medium, the possibility of releasing the carried polyphenol only in the desired target site may significantly improve the biological effect exerted by curcumin.”